# An Analysis of Students' Cognitive Bias in Experimental Activities Following a Lab Manual

**Seong Un Kim * and Il Ho Yang**

Institute of Nature Science Education,  Korea National University of Education, Chungbuk 28173, Korea**;** yih118@knue.ac.kr

* Correspondence: auul@naver.com

**Abstract:** The purpose of this study is to develop a thinking process model that reveals cognitive bias through analyzing students' cognitive biases in processing experimental manuals. Twenty-two college students participated in the study. During the "making electromagnets" experimental activity, we collected students' concurrent verbal protocols, gaze positions, and experimental behaviors. After the experiment, we collected their retrospective verbal protocols and ensured reliability by diversifying the data. The collected data were analyzed inductively using the grounded theory methodology. The results showed that four categories of paradigm (causal conditions, phenomena, interactions, and contextual conditions) and fifteen concepts were derived. Students displayed bias in following the manual instructions due to the influence of causal conditions. When embodying biased representations as workspace entities, biased responses come from the influence of contextual conditions. Therefore, these can be developed in consideration of causal and contextual conditions when developing a manual, thereby reducing cognitive bias among students, and ultimately helping them perform accurate experiments.

**Keywords:** cognitive bias; experimental activities; lab manual

## 1. Introduction

Scientific experimental activities are commonly used for learning science in schools across the globe [1]. Students' experience of experimental activities in elementary and secondary schools has been found to have a positive effect on the development of important abilities required for science education [2]. However, a number of prior studies have also expressed negative opinions about the effectiveness of experimental activities [3,4]. Therefore, various strategies have been studied to find and improve the causes of such a lack of effectiveness [2,5,6].

Millar et al. [7] noted that the primary cause of the ineffectiveness of experimental activities was inconsistency between what teachers intend for students to do versus what students actually did. Further, this inconsistency negatively affects what students are able to learn. Thus, in order to enhance the effectiveness of experimental activities, it must first be ensured that the manual is correctly followed and executed as intended by the teacher.

Often times, it is difficult for students to process the manual accurately. They may not understand the experimental procedure [8] or fail to accurately remember what they need to do [9,10]. This difficulty is said to stem from the lack of background knowledge or experience with the nature of the experimental activity that requires them to learn scientific concepts through new experiments [11].

Given this, there have been various efforts to supplement students' lack of knowledge and experience. Manuals are often improved by presenting diagrams or visualizations [12–14], providing preliminary information for processing the manual [9], and using clear and friendly language [15–

17]. These aforementioned studies focused on reducing the cognitive load arising from manual processing, making it easier for students to process manuals. However, easy handling of the manual is one thing and accurate processing is another. Therefore, reducing the cognitive load on students cannot be the fundamental solution.

This study assumes that students are unable to accurately handle their manuals because of their cognitive bias and focuses on the fact that students employ heuristic thinking in experimental activities. The human thought process is heuristic in nature—a person makes decisions unconsciously using only some of the information available [18]. Most people only focus on a few pieces of information and make decisions using empirical and intuitive reasoning based on a simplified model because they are not able to quickly identify and integrate the vast amount of information before them when making decisions [19]. Therefore, heuristic thinking sometimes hinders rational and logical decision making [18] and is likely to result in abnormal problem solving by cognitive bias [20].

In the field of science education, various studies have been conducted on confirming bias in scientific reasoning [21–28]. In the context of scientific reasoning, confirmation bias is considered to mislead explanations, blinded by the evidence that is inconsistent with beliefs, and disable logical and rational explanations based on evidence [29]. If there are many cognitive biases, even if the same information is encountered, problem solving may become irrational by selectively accepting it narrowly and applying an unscientific causal rate. In other words, eliminating cognitive bias can be one of the key strategies to handling experimental manuals correctly.

The purpose of this study is to develop a model that can explain the process and causality of cognitive bias by analyzing the cognitive bias that appears in students' manual processing. Based on these results, we thereafter discuss implications for how to accurately handle the manual and increase the effectiveness of the experimental activity.

## 2. Method

Cognitive bias is not readily apparent through behavior. In order to analyze cognitive bias, we need to collect students' internal thinking processes. In this study, the Think Aloud Method was used to collect the thinking process. Eye disease and eye vision were considered in collecting eye movement data. For theoretical saturation, we sampled students from various majors and backgrounds. To collect high-quality Think Aloud Method data, we recruited twenty-two national undergraduate students (5 males, 17 females) to participate in this study. We ensured that they were non-science majors and without any scientific expertise. Only data from 18 participants were analyzed; four participants were excluded as their cognitive biases were not evident in the data analysis.

The task was to develop experimental manuals that can allow students to observe natural phenomena and inductively learn science concepts by fully reflecting the experimental activities of elementary and secondary schools. The topic of the task was "making electromagnets". To accomplish the task, participants use nails and enamel wires to make electromagnets and observe the changes from opening and closing the switch, as they approach the dressmaker pins and compasses. Participants would ideally learn about the properties of electromagnets by observing the phenomena.

The task consisted of a title, preparation materials, experimental procedures, and a photograph of the experimental activity. The task involved reviewing the content validity of four science education experts who have experience in elementary and secondary science education and have eye tracking research experience as well. The evaluation criteria for the validity review consisted of four items, which included questions on whether the project correctly reflected the experimental activities of the elementary and secondary school field. 1. Is it a task that can improve conceptual understanding through the experimental activity? 2. Is it a task that participants in the study can perform individually? 3. Is the difficulty of the task appropriate for participants and unable to be resolved in an automated way? 4. Is the task to minimize the experimental error? The response for these items were to be rated on a 5-point Likert scale, and the final Context Validity Index (CVI) was 0.91.

Students' concurrent verbal protocols (CVP), gaze positions, experimental behaviors, and retrospective verbal protocols (RVPs) were collected. Reliability was ensured by diversifying the data. In-depth analysis of cognitive biases in experimental activities requires clues actually uttered by participants [30]. These clues appear in certain dynamic contexts, and researchers can use these clues to develop a description of the participant's thinking [31]. One useful clue here is to collect the CVP, which is a Think Aloud Method measure, of the participant during the experiment. This is an externalization of the contents of working memory during work [32], which is a continuous expression of the participant's ideas during work. The advantage of CVP collection is that people's expression of their thoughts that are spoken in the absence of any judgment makes it direct and automatic thought utterance without delay [33]. Because CVP content reflects thought as it is, it can be a means of constructing and validating cognitive processing theory [33].

Another clue is to collect their RVP, a Think Aloud Method measure about their experience, after the task has been completed. RVP collection can be used as a complement to the drawbacks of CVP collection. In fact, the cognitive process is faster than the verbal process, and hence, the research participants may actually be thinking more than during the verbal process, and because certain cognitive processes are unconscious, they may not be adequately expressed or may be totally ignored [34]. RVP collection is needed to offset these shortcomings and to complement the cognitive process represented by CVPs [35].

In addition, since the thinking process changes sensitively in response to changes in the working environment, there are very few clues that reveal the complex individual thinking process. Besides, there are many parts of the process where there is a high density of decisions to be taken, which is processed unconsciously. Therefore, it is difficult to analyze in detail what process actually occurs, and thus there are some thought processes that cannot be analyzed by the Think Aloud Method generally used in the study of cognitive processes. In this study, the eye tracking method was used to collect gaze positions to resolve these difficulties. Because gaze movements correspond to changes in situational awareness [36], gaze position can be used to infer perceptions of participants' situations in complex work situations [31].

The experimental process was as follows. Participants who entered the laboratory first performed the Think Aloud Method training for 15 to 20 minutes. Afterwards, the task was carried out while wearing the eye tracker, which collected CVPs, gaze positions, and experimental behavior. After the task was carried out, RVPs of the situation were collected using a gaze video recording the process of performing the task with an eye tracker. Gaze positions and CVPs were collected using Tobii Pro Glass 2 from Tobii, while RVPs and experimental behavior were collected by video camera.

In this study, the grounded theory methodology was used to theorize inductively based on actual phenomena to discover the conceptual meaning inherent in the phenomena. The data analysis follows the grounded theory methodology by Strauss and Corbin [37]. Thus, it uses a flexible approach to the study, rather than employing analytical and systematic procedures. In particular, the framework of the axial coding paradigm was intended not to be overly dependent on the structure presented by Strauss and Corbin [37], but rather to form a paradigm that would be useful for describing phenomena.

The experimental activity situation is very dynamic. The workspace changes rapidly and students are constantly required to make momentary changes. Therefore, to analyze the thought process closely, it is necessary to know what happens over time and what kind of thoughts occurred at that time. Participants express very different behaviors and thoughts as the environment changes. Therefore, the situation and the thoughts and behaviors of participants should be analyzed in an integrated way. Therefore, we transcribed gaze positions, RVPs, and experimental behavior according to the order of time around CVPs. The following table shows some of the transcriptions (Table 1).

**Table 1.** Example of a transcript. CVP, concurrent verbal protocols; RVP, retrospective verbal protocols.

| Time | Gaze position | Behavior | CVP | RVP |
|------|---------------|----------|-----|-----|

| 044 | Procedure no 4 End of enamel wire | Reading procedure no 4 | Then, using the sandpaper … the ends of the enamel wires? | |
|---|---|---|---|---|
| 045 | Sandpaper in the box Procedure no 4 Procedure no 2, 3 Electromagnet in the picture | With one hand, find the sandpaper in the box Touch the end of the enamel wire with the other hand | Does this strip off the sandpaper? Take off the cable sheath? | I again read the first sentence and the second sentence separately. Continuing, the cable sheath is a bit like this, and the cable sheath is a piece of rubber, that's called a cable sheath, right? I think the enamel wire was a bit confusing because I did not know what the enamel wire's cable sheath was. |
| 046 | Other supplies Procedure no 4 Sandpaper and enamel wire | Look for other supplies in the box Look again at the sandpaper and enamel wire | Is this not an enamel wire? There's only one wire. | I had this idea at first. Because there is nothing mentioned about the material, and I do not know anything about the material, I found this part confusing. At the moment, I began to think, did I start wrong from the beginning? As I kept thinking, I became a little convinced that this was an enamel wire, because there were no other supplies. But if there was another wire, I would have been confused. |
| 047 | Part of enamel wire rubbed with sandpaper | Fold the sandpaper outward, fold it in half, rub it with enamel wire | I do not know if the cable sheath will strip off or not. Ah! Stripped. | It looked like it was stripped because the color looked slightly worn. |

First, we repeatedly reviewed the transcription data and extracted the parts which showed cognitive bias. In this study, it is important to distinguish between cognitive bias and general problem-solving strategies. Thus, using Tversky and Kahneman's [38] definition of cognitive bias as "systematic deviation from normal behavior", we extracted thoughts and behaviors that differed from normal behavior. After that, the extracted data were reviewed to exclude general problem-solving strategies such as trial and error or means–ends analysis.

In order to increase the reliability of the cognitive bias extraction, we extracted cognitive bias from the data of three participants and analyzed it together with internal reviewer. Further discussion confirmed the cognitive bias. A total of 55 cognitive biases were extracted from 18 of the 22 participants. The data of 4 study participants were excluded from the analysis because no cognitive bias was extracted.

The extracted cognitive biases were analyzed according to the analysis stage of the grounded theory methodology. In the open coding stage, the concept is named through line-by-line coding. To extract similarities and differences between cognitively-biased experimental behavior and general experimental behavior, the extracted data were analyzed through a qualitative analysis strategy. Concepts were named and categorized through this process.

In the axial coding stage, categories were developed and linked to each other in relation to the occurrence of cognitive bias. The axial coding paradigms were modeled with a focus on "phenomena", the "causal condition" that contributes to cognitive bias, "interaction" with the workspace under cognitive bias, and the "contextual condition" that affects the type of interaction response.

In this study, we were aware that the experience and perspectives of researchers throughout the research process may act as preconceived ideas. To ensure the validity and reliability of the research, we have stated and described them in terms of the researcher's own reasoning. In addition, we discussed the naming and categorization of concepts in the coding process through the sharing of research contents with external consultants.

## 3. Results

In order to clearly understand the cognitive bias in the experimental activities, it is necessary to understand cognitive bias in comparison to normal manual processing, which does not show any cognitive bias. Normal manual processing can be derived by excluding conditions related to the occurrence of cognitive bias in the inductively modeled cognitive bias model.

The processing of experimental manuals requires the active use of knowledge and experience, as well as the information given in the manual [7,17]. In addition, the procedures written in the manual should be executed in the workspace to actively interact with the results being implemented. The normal manual processing model is shown in Figure 1.

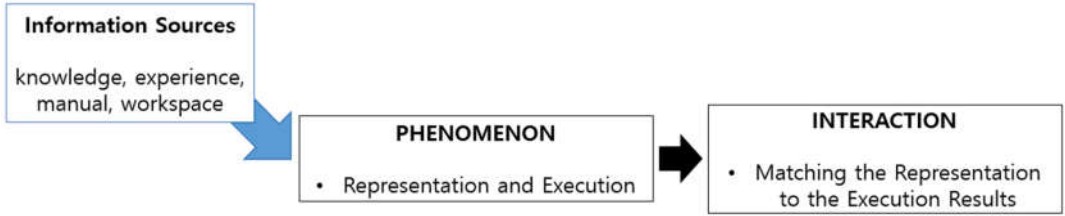

**Figure 1.** The normal manual processing model.

In the first stage of manual processing, information sources such as knowledge, experience, manual, and workspaces are used to represent the procedures presented in the manual and to execute what is represented in the workspace. Representation is the hypothetical, inner image or text of the mind, in which an entity is conceived [39].

Abrams and Millar [40] introduced representations as an internal process, in contrast to the execution of physically implemented tasks in the processing of the manual. Other studies [41,42] described the use of the manual and its subsequent maintenance of the procedure. In this study, the concept of "representation" in describing the thinking process is considered to be more structured and introduced.

The representations can be divided into two types depending on their content: procedural representation and outcome representation. A procedural representation is an internal representation of how a procedure is executed. It includes details such as time sequence, direction, and quantity. The processing of user manuals, such as furniture assembly, the process of mentally thinking how to assemble, is essential [43,44]. In processing the experimental manual, the procedure is also represented internally. In order to carry out the procedure correctly, it is essential that the procedural representation is as intended by the manual.

On the other hand, an outcome representation is an internal representation of the results that are expected to occur when the procedure is performed. In the processing of user manuals, it is important to carefully think about what would happen after the procedure [43,45]. In the processing of the experimental manual, the expected results from performing this procedure are represented internally. The outcome representation serves as a "standard" to confirm the successful execution of the procedure in comparison with the entities implemented in the workspace [43]. In this study, representation as a phenomenon appearing in manual processing was not clearly distinguished into procedure representation or outcome representation, because it is meaningless to distinguish clearly, but was used in analysis.

In the second stage of manual processing, we matched the representation and the execution result implemented in the workspace. This was achieved by analyzing the interaction with the

workspace, which compares and evaluates the workspace based on the representation. If the representation and the execution result do not match, the cause is primarily regarded as a problem in the workspace. Participants try to find and solve the cause in the workspace to match the representation with the execution result. However, if they do not find inconsistent causes in the workspace, they then look for the cause in the representation. If it is due to the incorrect representation of the procedure because they did not understand the manual correctly, they could go back to the initial step of reading the manual.

On the other hand, if the representation and the execution result match, the procedure is regarded as completed; the method of processing the next procedure is then repeated (representation and execution using the information source to match the representation and the execution result).

The thinking process model, which reflects cognitive bias, includes two additional conditions in the processing of the manual compared to normal manual processing. The cognitively-biased manual processing model derived from this study is shown in Figure 2.

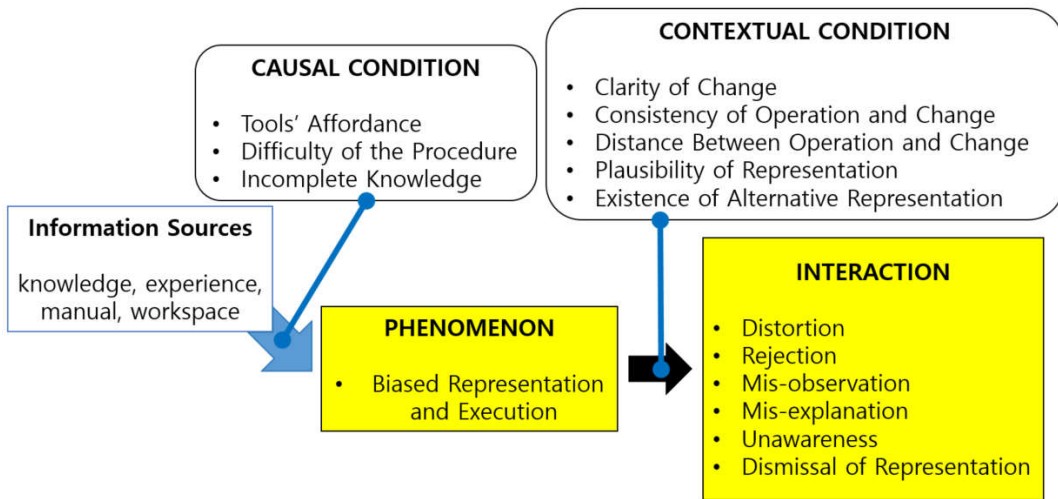

**Figure 2.** The axial coding paradigm model.

A cognitively-biased manual process is used to represent and implement the process using information sources, just as in normal manual processing. After that, it interacts with the results implemented in the workspace. The purpose of the interaction is to match the representation and the execution result of the workspace, as in a normal manual process. However, there is an important difference in that cognitive bias occurs in the thinking process.

The phenomena and interactions in yellow indicate the cognitively-biased part of the thinking process. In the process of representing and executing the procedure, which is the first step for processing the manual, biased representations are generated by causal conditions and executed accordingly. Subsequently, in the process of interacting with the workspace, which is the second step for processing the manual, contextual conditions affect the interaction and perceive the workspace as biased.

Table 2 describes the concepts and their explanations as inductively categorized from the data.

**Table 2.** Category and concept.

| Category | Concept | Explanation |
|---|---|---|
| Phenomenon | Biased Representation and Execution | Representing cognitive bias which differs from the manual's intent, and executing biased representation in workspaces |
| Causal Condition | Tool Affordance | Inducing a particular action using the apparent features of a tool |

| | Difficulty of the Procedure | Difficulty in understanding the full contents of the procedure |
|---|---|---|
| | Incomplete Knowledge | Wrong scientific concepts, related to previous experience |
| Interaction | Distortion | Being mistaken for observing expected results, even if there is no change in operation |
| | Rejection | Thinking of the cause of an experimental error as a methodological defect, not accepting the observation results |
| | Mis-Observation | Wrongly describing results due to erroneous observations |
| | Mis-Explanation | Wrongly describing results due to biased representation |
| | Unawareness | Failing to observe small changes |
| | Dismissal of Representation | Recognizing that a representation is biased, trying the representation again |
| Contextual Condition | Clarity of Change | Changes made by an operation are not ambiguous or obvious |
| | Consistency of Operation and Change | The same operations have the same variations; thus, the same changes are consistent with the operation |
| | Distance Between Operation and Change | The location and time of the operation and the location and time of the change are both close to and distant from each other |
| | Plausibility of Representation | Representation is detailed and can explain several things |
| | Existence of Alternative Representation | Alternative representations exist in addition to present representations |

## 3.1 Phenomena—Biased Representation and Execution

The first step in processing a manual is to represent and execute a procedure; to represent a procedure, given information sources must be used. At this time, the causal condition causes the biased representation. The biased representation is executed in the workspace and embodied as an external entity.

Participants do not easily notice that their representations are biased. It is not known whether the manual has been successfully handled as intended until the representation is realized in the workspace. This is because successful processing of procedures can be seen by checking the workspace to ensure that the representation and the result match. Checking whether the representation and the result match is determined based on biased representation. Therefore, it is difficult to judge whether the processing is successful because the criteria of judgment are biased. This shows that in manual processing, accurately representing the procedure is crucial for successful processing.

## 3.2 Causal Condition

The causal conditions that lead to biased representation in the processing of the information source do not appear to be clearly independent of each other. Students' internal conditions, such as "incomplete knowledge", affect external conditions such as "tool affordance" and "difficulties of the

procedure." Causal conditions are related to and interact with each other and serve as causes of biased representations.

### 3.2.1 Tool Affordance

Affordance, a term first proposed by Gibson [46], is present when a person detects a tool, automatically thinks of the tool's potential behavior, and uses it accordingly [47]. This means that features such as appearance and circumstances of certain tools naturally lead to specific actions.

The tools used in experimental activities have a distinctive appearance and a corresponding affordance. In the manual, demanding that "the electromagnet be brought close to the dressmaker pin with the switch closed", many participants showed that the dressmaker pin was brought closer to the electromagnet instead of the electromagnet being brought closer to the dressmaker pin. In response, we interpreted it to mean that it was empirically appropriate for participants to take small things (dressmaker pin) and bring these closer to them with larger items (electromagnet).

Another example showed that 10 participants did not take both poles of the electromagnet to the dressmaker pins but moved the enameled wire closer to the dressmaker pins. This seems to be due to the fact that the enamel wire was wrapped in the previous procedure of the manual and that it was thought to have a manipulative reaction in the wound part of the electromagnet. As such, the representation of a procedure is influenced by tool affordances.

### 3.2.2 Difficulty of the Procedure

Participants represent only what they understand when the content of the procedure is difficult to understand. Take for example the procedure "cut one end of the enamel line about 10 cm away and wind it 150 times in one direction." Many participants in the study questioned how and why the end of the enamel line should be left with a 10 cm measurement. Therefore, rather than agonizing to understand the meaning of the procedure, some participants in the study only represented and executed the understandable part: "Around and closely wound the enamel line 150 times in one direction." When they wound up all the enamel wires in a nail and connected them to the battery, they realized that the enamel they had to leave was the part that needed to be connected to the battery. It was only then that they realized they had not performed this part of the experiment correctly.

Understanding only a part of the process and flippant handling occur due to the availability bias, resulting in decisions that are based on information or ideas that can be quickly obtained from troubleshooting [48]. When it is difficult to understand the procedure, participants do not refrain from executing it but use only the information that is understandable to them. Procedures that they did not understand at the onset are thereafter understood throughout the process.

This processing appears to be due to the characteristics of the manual. In the manual, the previous procedure is logically intertwined with the following procedure. The state of the results of processing the previous procedure shall be the initial state of the following procedure, the reasons for handling the previous procedure and the expected results shall be associated with the following procedure, and so on. However, the incoherent connection between procedures is confusing in manual processing. In this process, participants infer and execute confusing procedures and reasons for implementing them.

### 3.2.3 Incomplete Knowledge

Background knowledge plays a very important role in manual processing. Familiar tasks that participants already have knowledge about can be processed more quickly and efficiently without difficulty in understanding the manual [42], even if there are errors in the instructions given [49]. On the other hand, if background knowledge is lacking, reasoning is needed to understand the manual. Given that this reasoning can be wrong [45], the effectiveness of manual processing largely depends on the amount of background knowledge a participant has.

Incomplete knowledge can likewise lead to biased representations. This may include preconceptions, misconceptions related to phenomena, and examples of past experiences. In this

study, incomplete knowledge of electric circuit, magnet, magnetization, current, and enameled wire, etc., was applied. For example, a participant who had the misconception that "a non-magnetic nail holds an object made of iron," expected, "because a nail has the clip attached, it is not an accurate result to attach the clip to either poles of the electromagnet when the switch is closed, and the clip should be attached to the enameled wire."

In addition, the past experiences of participants revealed in this study were similar experiment experiences, experiences with experimental tools, and science class experiences in old and impoverished elementary and middle schools. For example, past experiences of scraping off enameled sheaths with knives have led to misrepresentation, disregarding the procedure of "use sandpaper to peel off" as described in the manual.

### 3.3 Interaction

Since it is necessary to match the results of the workspaces with the representations to complete the processing of the procedures, we compared the workspaces with the representations after performing the biased representations. In this process, participants interacted with the workspace because it adjusted the workspace based on the representation. However, upon matching the result of the workspace based on the biased representation, the execution result was accepted as biased.

Execution with biased representations allows participants to observe what they expect and explain as expected. If the representation is biased, it can be seen that it is difficult to observe the execution results of the workspace accurately, to explain objectively, and to reject the biased representation.

Concepts in the interaction category can be distinguished by the type of interaction response with the workspace. The results are shown in Table 3.

**Table 3.** Taxonomy of interactions.

| | Was there a change by manipulation? | Is the change accurately observed? | Do they judge the change to be valid? | Do they explain the change? | Is the description of the change correct? |
|---|---|---|---|---|---|
| Distortion | X | X | O | X | - |
| Rejection | O or X | O | X | O | X |
| Mis-Observation | O | X | O | O | X |
| Mis-Explanation | O | O | O | O | X |
| Unawareness | O | X | X | X | - |
| Dismissal of Representation | O | O | X | O | O |

### 3.3.1 Distortion

Distortion is mistaking an observation for the expected result. Participant A closed the switch without the electrical circuit connected when the electromagnet was close to the pin. Thus, although the pin was not attached to the electromagnet, she stated that she observed that the pin was slightly attached and dropped. This reaction distorts the phenomena to conform with what was expected.

### 3.3.2 Rejection

Rejection is the distrust of the observable results in the workspace. Participant I observed the enameled wire wound in the electromagnet, close to the pin, and observed that the enameled wire and the pin did not stick together. The participant was expected to attach the pin to the electromagnet, and so he briefly showed an action to find the cause of the mismatch between the representation and the result, but failed to find the cause and explained that the experiment material—the pin—was not

suitable. These responses are interpreted as problems in manuals, experimental tools, or methodological flaws that fail to accept the results of the workspace as they are and do not correspond to the representations and the results.

### 3.3.3 Mis-Observation

Mis-observation is a different perception and explanation of the phenomenon due to incorrect operational observation. Participant Q brought the electromagnet closer to the compass to identify the properties of the electromagnet. However, it is not easy to observe changes in the direction of the compass needle by simply operating the electromagnet close to or away from the compass in one direction. Such incorrect operation causes the phenomenon to be observed and explained differently.

### 3.3.4 Mis-Explanation

Mis-explanation was an accurate observation of the phenomenon, but an incorrect description of what was observed. Participant V observed that the electromagnet was attached to the anode of the electromagnet, close to the pile of pin, and that the pin attached to the anode stretched downward to form an arch. However, he incorrectly explained this phenomenon as one in which an enameled wire wound around an electromagnet pulled a pin. Representation is used as a criterion for judging whether or not the result is consistent in the interaction with the workspace. The biased representation causes the phenomenon to be deflected and explained.

### 3.3.5 Unawareness

Unawareness is a small change in phenomenon that cannot be observed. Many participants determined whether the sheath had been peeled or not based on the color difference between the peeled part and the part that was not. However, the color difference between the peeled and unpeeled parts was so small that it was difficult to observe. In addition, it was difficult to perceive minute differences even when the representation was biased. These reactions are often manifested by experimental activities in which changes are not obvious but are aggravated by a lack of inquiry ability, such as observation and biased representation.

### 3.3.6 Dismissal of Representation

Dismissal of Representation is to observe and explain the phenomenon accurately, but to recognize that the current representation is biased and to dismiss it. Unlike the other five responses of "interaction", the dismissal of representation is an unbiased response. Recognizing different representations and executions than the intentions of the manual, the process returns to the representation of the corresponding procedure.

Chinn and Brewer [50] suggest that there are eight responses when students deal with anomalous data compared to their preconception. The interaction type of this study led to a similar response to anomalous data as in [50] (rejection, unawareness, dismissal of representation), but their response to the anomalous data was not similar (distortion, mis-observation, mis-explanation). This difference in response could be attributed to the differences in the procedural representations and preconceptions. Preconception, a product of empirical regularity, is a mental model of a phenomenon, and it cannot be easily changed. On the other hand, procedure representation in a simple experimental manual seems relatively easy to change.

In addition, prior belief strongly influences causal reasoning when evaluating evidence, as noted in several studies [51–54]. Likewise, in this study, when comparing the results of the workspace with the representations, the already biased representations have an absolute effect on the interaction with the results, making it impossible to observe the phenomenon accurately.

### *3.4 Contextual Condition*

Contextual conditions are conditions that affect the type of interaction response with the workspace. These conditions relate to the belief in representation and the characteristics of the observable performance of the workspace in its interaction with the results of the workspace.

The five concepts of contextual conditions presented below are not clearly independent of each other. The five concepts interact with each other and influence the types of responses that interact with the workspace.

### 3.4.1 Clarity of Change

The clarity of change is a matter of how clear the observable result is. Participants observe changes before and after the procedure is executed. At this point, the degree of change by manipulation must be clear at its own level to be perceived. If the changes are too small or ambiguous, it is difficult to observe the changes between before and after the operation, making it difficult to assess the successful processing of the procedures.

### 3.4.2 Consistency of Operation and Change

The consistency of the operation and the change is that the same change occurs in the same operation that participants perceive. If there is consistency in the relationship between operation and change, it is easy to infer the correlation between operation and change by performing operational observations, which can lead to inductive tendency. However, if the correlation between the operation and the change is not observed due to improper operational observations, the operation cannot be linked to the change.

### 3.4.3 Distance Between Operation and Change

The distance between the operation and the change is about how far apart the operation and the change are in space and time. The spatial distance is how far the resulting change takes place after manipulation; the temporal distance is how long it takes for this to happen after manipulation occurs. The smaller the spatial and temporal distance between manipulation and change, the more favorable it is to observe the phenomenon; the larger it is, the more disadvantageous it is to observe the phenomenon.

### 3.4.4 Plausibility of Representation

The plausibility of representation is the extent to which representation is a plausible explanation for the execution of a procedure. If the representation is logical and specific, participants perceive the representation as plausible. Logical and specific representations include not only procedural representations of how the procedures are to be carried out and outcome representations of expected outcomes, but also the purpose of implementing the procedures. Most importantly, the more plausible the representation is, the more confident the participant is. They think the representation is plausible because it accurately reflects the intent of the manual. However, if the representation is biased and there is high confidence in the representation, it negatively affects the processing of the manual. If the representation and the result do not match because of the biased representation, they consider the problem to be the workspace, rather than thinking that the cause is a matter of representation. Instead, they look for the cause of the discrepancy in the workspace. In other words, if the representation is likely, there is high confidence even if the representation is biased. Therefore, they interpret the result as a frame of representation. On the other hand, if the representation is not likely, it is easy to dismiss the representation because of the lack of confidence in the representation.

### 3.4.5 Existence of Alternative Representation

The existence of alternative representations is to represent two or more representations in the early stages of representing procedures. If the procedure can be understood in two ways, one of the two meanings is chosen and represented, so there are alternative representations that differ from the

current representation. If the participant has an alternative representation, it can be easily rejected if the current representation is inconsistent with the outcome.

## 4. Discussion

School experimental activities are inherently complex and ambiguous. Therefore, teachers can control students' experimental activities as intended by providing them detailed procedures through manuals [55]. Previous studies have increased the readability of experimental manuals [56] and structured them to make it easier for students to follow [12,13,16,55]. These high levels of structuralization and detailed guidelines make it easy for students to learn scientific concepts. However, this strategy makes experimental activities meaningless because it makes it difficult to understand the nature of science, scientific practice, and scientific knowledge that students need to learn through scientific experiment activities [2,55]. It prevents students from high-level thinking, and they continue to stay in the "follow the recipe experiment" [5].

As such, there is conflict between the two aspects of improving the effectiveness of experimental activities. There is the dilemma of what to emphasize between the structure of the experimental manual and the high-level thinking experience. To easily follow the experiment activities and to increase the effectiveness of the experiment activities, the experimental manual should be structured. However, if it is highly structured, the effectiveness of learning through the experiment activities is lowered. How do we find the right balance to ride both horses?

This study therefore focuses on how to make students follow the experimental manual accurately, based on the structure of the experimental manual. Analyzing the cognitive bias, which leads to inaccuracies in following the lab manual, provides a completely different view of the effectiveness of the experiment. By removing the part of the experimental manual that may generate cognitive bias by reflecting on the results of the research, it is imperative that we need to construct a manual that students can follow accurately.

The results of this study show that some characteristics are different from the results of previous studies modeling experimental manual processing. Several previous studies emphasize declarative and procedural knowledge [7,17,57–59]. A successful process is closely related to knowledge. In this study too, knowledge related to tasks, and incomplete knowledge, affects cognitive bias. However, in this study, students make decisions using intuitive heuristic strategies rather than rational and logical processing with knowledge and information. In this process, representation can be perceived to be cognitively biased. In particular, tool affordance was derived as one of the causal conditions affecting manual processing. To process the manual correctly, in addition to the background knowledge related to the task, consideration should be given to the affordances of tools empirically and inherently possessed by humans. To prevent cognitive bias, it is necessary to identify tool affordances that adversely affect students.

In addition, previous studies emphasize the classification and selection of information for processing experimental manuals. This is because "noise," which is irrelevant to the performance of experiments, hinders performance, slows it down, and draws irrelevant schemas. To not fail the manual process, noise must be reduced through manual refinement, and students must distinguish between signal and noise in the search for information, discard noise, and focus on the signal [17,57,59]. However, it is not necessary to focus on dividing information from the outside into signal and noise. More importantly, because the signal is transformed into noise or noise into signal, it will interfere with the correct processing, and hence, the manual should be devised to reduce this distortion.

## 5. Educational Implications

Based on the results of this study, we can derive ways to prevent cognitive bias in experimental activities. In manual processing, cognitive bias occurs at points where the procedure is represented and at points where it interacts with the results of the workspace by executing it according to the biased representation. Therefore, to prevent cognitive bias, it is necessary to adjust causal and contextual conditions that affect the cognitive bias of two points.

In this study, three concepts were derived from causal conditions. The first is "tool affordances", in which it is found that the characteristic appearance of the experimental tool induces specific behaviors, which then cause a biased representation. Therefore, in order to prevent this, the manual should be prepared considering the induced behavior recognized by the students so as not to deviate from the exact experiment. Participants' experiments revealed that the electromagnets, enamel wires, pins, and compasses, which are the experimental tools of this study, have their own affordances. The trend in tool affordance in this study was shown to have been affected by background knowledge and experience, but further research is needed to understand them in depth.

The second is the "difficulty of the procedure". If it is difficult to understand the entirety of a particular procedure, people tend to use only some of the information that is understandable in the procedure to represent and execute it. The use of only select information will likewise cause a biased representation. Therefore, in order to prevent this, the procedure should be configured to facilitate an understanding of the whole process. The results of the previous procedure should be logically constructed so that the status can be the initial state of the next procedure, and procedures that are difficult to understand should be accompanied by aids like adding visual materials [12].

The third is "incomplete knowledge". A lack of background knowledge and experience affects one's understanding of the procedure. What is more serious is the use of highly available inaccurate knowledge and experience to create biased representations. Students should have no experience with the manual in the new lab but should be aware that they have similar experiences and preconceptions. Background knowledge that can lead to biased representations can be prevented by understanding what knowledge and experience students have with regard to the manual processing work.

In addition, this study derives five concepts as contextual conditions. Among these, the concepts of "clarity of change", "consistency of operation and change", and "distance between operation and change" are unrelated to students' beliefs but are connected to the characteristics of experimental activities. Thus, they can be adjusted and controlled at the manual level.

The first is "clarity of change". Among the interaction reactions, "unawareness" is a biased reaction in which changes are not observed because of the slight change in phenomena. Therefore, if the change caused by the manipulation is small due to the nature of the experiment activity, the manual can explicitly describe the small change and suggest observational viewpoints to help students observe the changes more in depth.

The second is "consistency of operation and change". In order to generalize the results, it is necessary to induce it based on the consistency of manipulation and change. This requires that the same change be observed in the same operation. However, erroneous operational observations make it difficult to generalize the results because they prevent the same change from being observed. In order to prevent erroneous operational observations, guiding observation points or guided operational observations can help in observing the phenomenon more accurately.

The third is the "distance between operation and change". If the temporal and spatial distance between operation and change is far, it seems difficult to establish causality in observing and interpreting the phenomenon. Therefore, if the temporal and spatial distance between manipulation and change differ due to the nature of the experiment, it should be specified or guided by the observation point, so that focus can be placed on the necessary part.

Ultimately, accurate manual processing without cognitive bias will contribute to the effectiveness of the experiment. However, guiding the manual to be handled correctly, as claimed by science educators, there are concerns about inducing experimental activities in the form of a "simple cookbook" that impedes the effectiveness of experimental activities [3,4,60]. In other words, efforts to increase the accuracy of experimental activities by reducing cognitive bias appear to conflict with each other in their efforts to increase the effectiveness of experimental activities.

To enhance the effectiveness of experimental activities, the opportunity to participate in the research process of scientists should allow them to experience high-level thinking [2]. Therefore, to reduce the cognitive bias and increase the effectiveness of the experiment, one should not simply follow the manual correctly. We must provide opportunities for reasoning at higher levels. To do so,

the procedure needs to be presented separately as procedures that must be followed accurately and procedures that require scientific reasoning. Procedures that require accurate execution in the production of evidence should not be cognitively biased. Instead, procedures should be constructed in consideration of causal and contextual conditions. Conversely, procedures that evaluate evidence and require in-depth observation should provide an experience of observing a phenomenon as it is by reducing the portion of the expected outcome that can be inferred.

## 6. Conclusion

Human beings use a heuristic strategy in problem-solving situations that lack information, thereby leading to biased thinking. It is necessary to focus on cognitive bias to identify and correct the cause of abnormal thinking. Cognitive bias is difficult to extract because it is handled almost automatically in dynamic situations. Therefore, this study attempted a new method for collecting and analyzing the thinking process. In this study, CVPs, eye movement data, RVPs, and experimental behavior were collected to diversify the data. The model was developed by inductive categorization by applying the grounded theory methodology.

Categories and concepts that emerged from cognitively-biased manual processing were derived. Manual processing consists of categories such as "phenomena" that represent and execute cognitive biases, "interactions" that match the representations and results of the workspace, "causal conditions" that affect the occurrence of cognitive bias, and "contextual conditions" that affect the type of interaction.

Cognitive bias occurs at points representing procedures and at points interacting with the workspace by executing representations. Therefore, to prevent cognitive bias, the manual should be developed considering the causal and contextual conditions that affect the cognitive bias of two points.

In addition, in order to increase the effectiveness of experimental activities, it is necessary to provide an opportunity to follow the procedure correctly and to observe the phenomenon as it is and to make high-level reasoning. To do this, the manual must be developed in two parts: one that must be followed exactly and one that requires scientific reasoning.

**Author Contributions:** Conceptualization, S.K. and I.Y.; methodology, S.K.; software, S.K.; validation, S.K., and I.Y.; formal analysis, S.K.; investigation, S.K.; resources, S.K.; data curation, S.K.; writing—original draft preparation, S.K. and I.Y.; writing—review and editing, S.K. and I.Y.; visualization, S.K.; supervision, S.K.; project administration, S.K. funding acquisition, S.K.. All authors have read and agreed to the published version of the manuscript.

**Funding:** This research was funded by the Ministry of Education of the Republic of Korea and the National Research Foundation of Korea (NRF-2018S1A5B5A02034210).

**Conflicts of Interest:** The authors declare no conflict of interest.

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
