# Peer review of "An Analysis of Students’ Cognitive Bias in Experimental Activities Following a Lab Manual"

_education, doi:10.3390/educsci10030080_

Round 1
Reviewer 1 Report
The purpose of the manuscript is to develop a thinking process model that reveals cognitive 4 bias through analyzing students' cognitive biases in processing experimental manuals. Only 20 references are included in the introduction section. It would be desirable to include more similar studies, to reinforce its scientific soundness. A strong point about the introduction is that authors include the purpose of the study in this section.
In the method section authors have not specified the selection process for the 22 participants.
In the line 72, authors say that "The content validity of four science education experts was also reviewed", but How was this process? Was there a rubric? Why were these experts selected?
Results are well explained, but there are not discussion section. Instead, authors include a section on implications. However, including a discussion section would increase its quality. Both the presentation of results and their reliability in relation to similar studies are important.
Conclusion section is correct.
Author Response
Thank you for your kind reply. We tried to correct all the edits you requested.
The purpose of the manuscript is to develop a thinking process model that reveals cognitive 4 bias through analyzing students' cognitive biases in processing experimental manuals. Only 20 references are included in the introduction section. It would be desirable to include more similar studies, to reinforce its scientific soundness. A strong point about the introduction is that authors include the purpose of the study in this section.
-> The introduction introduces the research on cognitive bias in the field of science education and emphasizes the necessity of looking at the perspective of cognitive bias in order to accurately process the experimental manual.
In the method section authors have not specified the selection process for the 22 participants.
-> Participants were recruited to college students of various majors and backgrounds for the theoretical saturation of research, and for four-year college students for high-level thinking.
In the line 72, authors say that "The content validity of four science education experts was also reviewed", but How was this process? Was there a rubric? Why were these experts selected?
-> Four scientific experts described their expertise as a validity reviewer (eye tracking research experience, primary and secondary education experience). And four questions are described for validity review.
Results are well explained, but there are not discussion section. Instead, authors include a section on implications. However, including a discussion section would increase its quality. Both the presentation of results and their reliability in relation to similar studies are important.
-> We described the discussion. The implications of this study and the comparison of previous studies on experimental manual processing were discussed.
Conclusion section is correct.
Reviewer 2 Report
Dear authors,
First, thank you very for submitting your paper to Education Sciences.
In this paper, 18 students’ cognitive biases in processing experimental manuals were analyzed. Grounded theory methodology was used to analyze the results.
In my opinion, the main weakness of the work concerns the use of scientific literature. The most current reference is from the year 2016. It would be interesting to enrich the article with more current references.
Introduction:
In my opinion the introduction is adequate, the main goals of the research are described.
Methods:
Please, describe further CVI.
Include some reference that use “Think Aloud Method”. Describe further this methodology.
You should further describe the protocol for the interpretation of eye track.
Results and discussion:
Sections 3.3. and 3.4. need further discussion. I think they should be compared with previously published work.
Minor revisions:
- In lines 37 y 39. References should be [12-14] and [15-17]. Please revise across the manuscript.
- In line 67, “making electromagnets.” should be typed “making electromagnets”.. Please, revise across de manuscript (lines 407, 412 and 418 present the same problem)
- Remove tab in table 1 and 2.
- Year should be included in reference 10.
I hope that my comments and suggestions should be useful for improving the artcile.
Author Response
Thank you for your kind reply. We tried to correct all the edits you requested.
In my opinion, the main weakness of the work concerns the use of scientific literature. The most current reference is from the year 2016. It would be interesting to enrich the article with more current references.
-> Your advice is a good point. However, recent studies on experimental manual processing have not been conducted. In addition, since this study is a basic research for developing new models in areas where there is a lack of previous research, we used evidence theory method. We also tried to make up as rich as you pointed out.
In my opinion the introduction is adequate, the main goals of the research are described.
-> The introduction introduces the research on cognitive bias in the field of science education and emphasizes the necessity of looking at the perspective of cognitive bias in order to accurately process the experimental manual.
Please, describe further CVI.
-> We described in detail the process of obtaining expert validity to explain CVI. The experts have expertise in a particular area and have detailed four questions for evaluation.
Include some reference that use “Think Aloud Method”. Describe further this methodology.
-> We describe the think aloud method in detail, separated into CVP and RVP.
You should further describe the protocol for the interpretation of eye track.
-> We have described the eye tracking method in more detail and provided examples of transcriptions to help readers understand the initial analysis.
Sections 3.3. and 3.4. need further discussion. I think they should be compared with previously published work.
-> We described the discussion. The implications of this study and the comparison of previous studies on experimental manual processing were discussed.
Minor revisions:
- In lines 37 y 39. References should be [12-14] and [15-17]. Please revise across the manuscript.
- In line 67, “making electromagnets.” should be typed “making electromagnets”.. Please, revise across de manuscript (lines 407, 412 and 418 present the same problem)
- Remove tab in table 1 and 2.
- Year should be included in reference 10.
-> We corrected all of the points you pointed out and checked it several times.
Round 2
Reviewer 1 Report
The authors have considered the comments made in the previous review and have improved the overall quality of the manuscript.
Author Response
The authors have considered the comments made in the previous review and have improved the overall quality of the manuscript.
-> Thank you for your consideration in helping us revise the manuscript.

Reviewer 2 Report
Dear authors,
Thank you for submitting the new version of your paper. In my opinions, this paper has been improved.
In my opinion this paper could be published with minor corrections:
- Figure captions and table captions are not according journal instructions.
- A space is needed after CVI (line 92).
- A space is needed before [ (line 131).
- Please, remove blank space in line 140. Revise others 142, 179, etc.
- In my opinion, authors should include some recent papers about Cognitive Bias published in literature.
Best regards,
Author Response
In my opinion this paper could be published with minor corrections:
-> Thank you for your consideration in helping us revise the manuscript.
Figure captions and table captions are not according journal instructions.
-> We modified it according to your advice.
A space is needed after CVI (line 92).
-> We modified it according to your advice. And we checked for similar errors in other parts of the manuscript.
A space is needed before [ (line 131).
-> We modified it according to your advice. And we checked for similar errors in other parts of the manuscript.
Please, remove blank space in line 140. Revise others 142, 179, etc.
-> We modified it according to your advice. We fixed it by referring to the original tamplate to see if there were any errors in other parts.
In my opinion, authors should include some recent papers about Cognitive Bias published in literature.
-> As your advice, we added a recent article(van der Graaf et al., 2016; Stover, 2016; Liu & Lawrenz, 2018; Varma et al., 2018) on Cognitive Bias.
